# Different miRNA Profiles in Plasma Derived Small and Large Extracellular Vesicles from Patients with Neurodegenerative Diseases

**DOI:** 10.3390/ijms22052737

**Published:** 2021-03-08

**Authors:** Daisy Sproviero, Stella Gagliardi, Susanna Zucca, Maddalena Arigoni, Marta Giannini, Maria Garofalo, Martina Olivero, Michela Dell’Orco, Orietta Pansarasa, Stefano Bernuzzi, Micol Avenali, Matteo Cotta Ramusino, Luca Diamanti, Brigida Minafra, Giulia Perini, Roberta Zangaglia, Alfredo Costa, Mauro Ceroni, Nora I. Perrone-Bizzozero, Raffaele A. Calogero, Cristina Cereda

**Affiliations:** 1Genomic and post-Genomic Unit, IRCCS Mondino Foundation, 27100 Pavia, Italy; daisy.sproviero@mondino.it (D.S.); stella.gagliardi@mondino.it (S.G.); szucca@engenome.com (S.Z.); marta.giannini@mondino.it (M.G.); maria.garofalo@mondino.it (M.G.); orietta.pansarasa@mondino.it (O.P.); 2EnGenome SRL, 27100 Pavia, Italy; 3Department of Molecular Biotechnology and Health Sciences, Bioinformatics and Genomics Unit, University of Turin, 10126 Turin, Italy; maddalena.arigoni@unito.it (M.A.); raffaele.calogero@unito.it (R.A.C.); 4Department of Brain and Behavioral Sciences, University of Pavia, 27100 Pavia, Italy; alfredo.costa@mondino.it; 5Department of Biology and Biotechnology (“L. Spallanzani”), University of Pavia, 27100 Pavia, Italy; 6Department of Oncology, University of Turin, 10060 Turin, Italy; martina.olivero@ircc.it; 7Departments of Neurosciences, University of New Mexico School of Medicine, Albuquerque, NM 87131, USA; micheladellorco@salud.unm.edu; 8Immunohematological and Transfusional Service and Centre of Transplantation Immunology, IRCCS “San Matteo Foundation”, 27100 Pavia, Italy; s.bernuzzi@sanmatteo.pv.it; 9Neurorehabilitation Unit, IRCCS Mondino Foundation, 27100 Pavia, Italy; micol.avenali@mondino.it; 10Unit of Behavioral Neurology, IRCCS Mondino Foundation, 27100 Pavia, Italy; matteo.cottaramusino@mondino.it (M.C.R.); giulia.perini@mondino.it (G.P.); mauro.ceroni@mondino.it (M.C.); 11Neuro-Oncology Unit, IRCCS Mondino Foundation, 27100 Pavia, Italy; luca.diamanti@mondino.it; 12Parkinson Unit and Movement Disorders Mondino Foundation IRCCS, 27100 Pavia, Italy; brigida.minafra@mondino.it (B.M.); roberta.zangaglia@mondino.it (R.Z.); 13Departments of Neurosciences and Psychiatry and Behavioral Health, University of New Mexico School of Medicine, Albuquerque, NM 87131, USA; NBizzozero@salud.unm.edu

**Keywords:** Alzheimer’s disease, Parkinson’s disease, amyotrophic lateral sclerosis, frontotemporal dementia, large extracellular vesicles, small extracellular vesicles, RNA-seq, miRNAs

## Abstract

Identifying biomarkers is essential for early diagnosis of neurodegenerative diseases (NDs). Large (LEVs) and small extracellular vesicles (SEVs) are extracellular vesicles (EVs) of different sizes and biological functions transported in blood and they may be valid biomarkers for NDs. The aim of our study was to investigate common and different miRNA signatures in plasma derived LEVs and SEVs of Alzheimer’s disease (AD), Parkinson’s disease (PD), Amyotrophic Lateral Sclerosis (ALS) and Fronto-Temporal Dementia (FTD) patients. LEVs and SEVs were isolated from plasma of patients and healthy volunteers (CTR) by filtration and differential centrifugation and RNA was extracted. Small RNAs libraries were carried out by Next Generation Sequencing (NGS). MiRNAs discriminate all NDs diseases from CTRs and they can provide a signature for each NDs. Common enriched pathways for SEVs were instead linked to ubiquitin mediated proteolysis and Toll-like receptor signaling pathways and for LEVs to neurotrophin signaling and Glycosphingolipid biosynthesis pathway. LEVs and SEVs are involved in different pathways and this might give a specificity to their role in the spreading of the disease. The study of common and different miRNAs transported by LEVs and SEVs can be of great interest for biomarker discovery and for pathogenesis studies in neurodegeneration.

## 1. Introduction

Neurodegenerative disorders (NDs) are a group of diseases characterized by loss of neurons within the brain and/or spinal cord [1]. They include Alzheimer’s disease (AD), Parkinson’s disease (PD), Amyotrophic Lateral Sclerosis (ALS) and Fronto-Temporal Dementia (FTD) [2]. Each of these disorders is characterized by specific features, both clinical and pathological involving characteristic central nervous regions [3].

AD is characterized by the extracellular accumulation of beta amyloid (Aβ) peptide detectable as Aβ plaques and intracellular Tau protein in the parenchyma and cerebrovasculature of the brain [4]. In PD degeneration of dopaminergic pigmented neurons in the substantia nigra (SN) and accumulation of α-synuclein protein, which is the main component of Lewy bodies (LBs), are the main features [5]. ALS is a disease characterized by motor neurons death and one of the main pathological hallmarks is given by specific alterations of SOD1 [6,7,8,9,10], and aggregation of TDP-43 [11]. In FTD, there is a deregulation of RNA-binding proteins (RBPs) and aggregation of proteins in the frontal and temporal lobes with microvacuolation, neuronal loss and astrocytic gliosis [12]. TDP-43 and Tau aggregates are hallmarks of FTD [13].

The aberrant RNA metabolism processing converges as a common factor in the pathogenesis of these diseases [14,15]. Abnormal RNA metabolism is associated with disease-specific alterations in RNA-binding proteins (RBPs), and in non-coding RNAs, such as microRNAs (miRNA), transfer RNAs (tRNA) and long-noncoding RNAs (lncRNA) [16].

Several common and specific mechanisms of NDs are described in the literature, however, no biomarkers are available to identify the onset, progression and comorbidity of those diseases. Brain cells release extracellular vesicles (EVs), which can go through the brain barrier [17,18] and blood derived EVs can be used also to monitor disease processes occurring in the brain [19,20,21]. EVs, spherical vesicles heterogeneous in size (30 nm–1 µm in diameter) are transporters of receptors, bioactive lipids, proteins, and nucleic acids, such as mRNAs, lncRNAs and miRNAs [22,23,24,25]. EVs are classified as: exosomes (EXOs), microvesicles (MVs), and apoptotic bodies [22,23]. EXOs are secreted membrane vesicles (approximately 30–150 nm in diameter) formed intracellularly and released from exocytosis of multivesicular bodies, whereas apoptotic bodies (approximately 1000–4000 nm in diameter) are released by dying cells. MVs (approximately 100–1000 nm in diameter) are shed from cells by outward protrusion (or budding) of a plasma membrane followed by fission of their membrane stalk [22,23]. However, the guidelines of the International Society for the study of Extracellular Vesicle (ISEV) released in 2018 declare that MVs and EXOs cannot be distinguished on a particular biogenesis pathway and so they can be distinguished in small extracellular vesicles (SEVs) (30–130 nm) and large extracellular vesicles (LEVs) (130–1000 nm) mainly on their size [24].

Several studies on the role of EVs in NDs are available in the literature. Some studies have examined miRNAs and RNAs in EVs isolated from cultured cell media from the Central Nervous System (CNS) cells (e.g., neurons, astrocytes, microglia, and oligodendrocytes) and few have examined miRNAs in EVs in plasma of AD, PD and ALS [25,26,27]. However, none of these studies considered the differences between SEVs from LEVs. We previously described in ALS that SEVs and LEVs in plasma are different in dimensions and for loading of some pathological proteins for ALS (SOD1, TDP-43, p-TDP-43, and FUS) and lipids [28,29,30]. In this paper, we have investigated the miRNA cargo of EVs derived from plasma of patients affected by four neurodegenerative diseases (AD, PD, ALS and FTD). The aim was to identify common and specific small RNAs between the two subpopulation of EVs in the same NDs disease and in the four diseases in order to identify new biomarkers.

## 2. Results

### 2.1. miRNAs Selectively Traffic into SEVs and LEVs

We previously demonstrated significant differences between LEVs and SEVs derived from plasma for dimension, markers, protein loading (see Appendix A) [28,29,30]. Cellular miRNAs can selectively traffic into LEVs and SEVs, so we first identified differentially expressed miRNAs (DE miRNAs) in SEVs and LEVs among the four groups of patients (AD, PD, ALS, FTD) and the healthy controls (CTRs) (Table 1, Appendix A). We then moved to investigate the number of different and common deregulated miRNAs that sort into SEVs and LEVs in the same disease. In AD, of the 33 miRNAs found in SEVs and 13 in LEVs, 6 distribute to both (Figure 1a), 4 up-regulated and 2 down-regulated. In FTD, of the 88 miRNAs in SEVs and 130 in LEVs, 34 were in common, 32 up-regulated and 2 down-regulated (Figure 1b). Concerning miRNAs in ALS, of the 109 miRNAs in SEVs and 197 in LEVs, 67 were in common, (Figure 1c), 45 up-regulated and 22 down-regulated. In PD, of the 104 miRNAs found in SEVs and 109 in LEVs, 34 distribute to both (Figure 1d), 30 up-regulated and 4 down-regulated. The percentage of common miRNAs between SEVs and LEVs are shown in Table 2.

### 2.2. miRNAs Expression Profiles and Common Pathways in SEVs and LEVs of NDs

miRNAs detected as differentially expressed in SEVs were pooled together and analyzed by principal component analysis (PCA) (Figure 2a). The same approach was applied to LEVs.

As shown in Figure 2a, deregulated miRNAs in SEVs of the four NDs did not overlap with the CTRs (orange). Interestingly, clusters related to ALS (light green) and AD (dark green) patients could be identified in separate components, while PD (violet) overlapped with some ALS specimens. The miRNA cargo of SEVs were well differentiated in all four diseases.

The PCA of the differential expressed miRNAs detected in LEVs did not provide instead a clear separation of the four diseases (Figure 2b).

However, miRNAs in LEVs of ALS patients could be distinguished from CTRs and from the other three diseases, with some overlap with FTD.

In order to detect common miRNAs of the four diseases, we calculated the intersection of deregulated miRNAs compared to CTRs. We observed that 6 miRNAs were in common among the four diseases in SEVs (hsa-miR-133a-3, hsa-miR-543, hsa-miR-4451, hsa-miR-6889-5p, hsa-miR-4781-3p, hsa-miR-323b-3p) (Figure 3a) and 7 miRNAs (hsa-miR-1262, hsa-miR-3152-3p, hsa-miR-7856-5p, hsa-miR-365a-5p, hsa-miR-4433b-5p, hsa-miR-6068, hsa-miR-767-3p) in LEVs (Figure 3b). Enriched pathways found with MiRWalk included Ubiquitin mediated proteolysis, MAPK signaling pathway, Toll-like receptor signaling pathways for SEVs and Neurotrophin signaling pathway, MAPK signaling pathway, Glycosphingolipid biosynthesis, Ras signaling pathway for LEVs (Figure 3a,b).

### 2.3. Specific miRNAs Pathway Analysis in SEVs of NDs

Enriched pathways targeted by the miRNAs that were differentially expressed between disease samples and CTRs were investigated by MiRWalk (Appendix A).

Reactome and Gene Ontology (GO) are listed in Appendix A.

In SEVs from ALS patients, Reactome analysis showed miRNAs involved in MECP2 (methyl CpG binding protein 2) expression and activity, Intracellular signaling by second messengers and negative regulators of DDX58/IFIH1. IFIH1 and DDX58 encode retinoic acid-inducible gene I (RIG-I) (Appendix A), cytosolic pattern recognition receptors function in viral RNA detection initiating an innate immune response through independent pathways that promote interferon expression [31]. Ingenuity pathway analysis (Figure 4a) confirmed regulation of the immune system by highlighting how miR31-5p and miR615-5p control CXCL8, C-X-C Motif Chemokine Ligand 8 activated in viral infections and miR584-5p/miR148-5p control TMEM9, which enhances production of proinflammatory cytokines induced by TNF, IL1B, and TLR ligands [31]. GO-BPs found negative regulation of Ras protein signal transduction and cell proliferation in forebrain.

In SEVs from FTD patients Reactome classified deregulated miRNAs in MyD88 cascade initiated on plasma membrane, diseases of signal transduction and axon guidance (Appendix A). GO-BPs classified deregulated miRNAs in SEVs from FTD in pathway release of cytochrome c from mitochondria, mitotic G1 DNA damage checkpoint and DNA damage response, signal transduction by p53 class mediator resulting in cell cycle arrest. IPA analysis (Figure 4b) confirmed involvement in MyD88 cascade (Adapter protein involved in the Toll-like receptor and IL-1 receptor signaling pathway in the innate immune response) and DNA damage response by regulation of miR146a-5p on RAD54, involved in homologous recombination and repair of DNA and on IL1F10 and TLR10 [32,33] (Figure 4b).

In SEVS from PD patients Reactome analysis identified signaling by TGF-beta family members, SLC-mediated transmembrane transport (regulation of GRIA1, subunit of Glutamate receptors, by miR-132-3p, Metabolism of lipids, MyD88 cascade initiated on plasma membrane (involvement of JAG1 and TNF regulated by miR-34-Sup IPA), while GeneOntology BP fibroblast migration, mRNA polyadenylation and positive regulation of gene silencing by miRNA (Figure 4c).

For AD, there were very few deregulated miRNAs to be classified in pathways.

### 2.4. Specific miRNAs Pathway Analysis in LEVs of NDs

Reactome underlined the role of miRNAs in LEVs (Appendix A): (1) from ALS patients in intracellular signaling by second messengers (DAG, cAMP, cGMP, IP3, Ca2+ and phosphatidylinositols), signaling by TGF-beta family members, MyD88 cascade initiated on plasma membrane and metabolism of lipids; (2) from PD patients in intracellular signaling by second messengers, SLC-mediated transmembrane transport (solute carrier superfamily), some of which mediate neurotransmitter uptake in the CNS and peripheral nervous system (PNS) and metabolism of lipids. No significative pathways were detected for miRNAs of LEVs from FTD patients.

Gene ontology analysis found also pathways involved: (1) in regulation of G0 to G1 transition in LEVs from ALS patients; (2) in regulation of type I interferon-mediated signaling pathway and positive regulation of gene and posttranscriptional silencing in LEVs from FTD patients; (3) protein insertion into membrane, fibroblast apoptotic process, regulation of steroid biosynthetic process in LEVs from PD patients. Pathways are reported below in order of most significant *p* values and further details are shown in Appendix A. IPA analysis confirmed the pathway found by both Reactome and Gene Ontology. (1) for LEVs from ALS regulation of p53 by mir-379, 378, 584-5p, 1207 emerged. hsa-miR-199a-5p and hsa-miR-329a-5p regulate LCN2, lipocalin2, inducible factor secreted by reactive astrocytes in transgenic rats with neuronal expression of mutant human TDP-43 or RNA-binding protein FUS and that is selectively toxic to neurons [34] (Figure 5a); (2) for LEVs from FTD patients regulation of genes like TP53 by miR-296 and RNA Polymerase II by miR-615 as well as Cyclin E regulated miR-615-5p, that might be part of gene expression and regulation of apoptosis [35] (Figure 5b); (3) for LEVs from PD patients regulation of ADAM9 (ADAM Metallopeptidase Domain 9), important in mediating cell–cell and cell–matrix interactions, by up-regulation of miR-291, miRNA that regulates cell proliferation and resolvin, a metabolic byproduct of omega-3 fatty acids, regulated by miR-302 [36,37] (Figure 5c).

## 3. Discussion

Identifying biomarkers is essential for early diagnosis of NDs. Extracellular vesicles (SEVs and LEVs) transported in blood might play this role. In this study, we have analyzed SEVs and LEVs cargo in order to detect specific and common miRNAs acting as novel, easily accessible biomarkers for AD, PD, ALS and FTD.

We first compared miRNA expression profiles of SEVs and LEVs in the same disease. We found a variable range of overlap between LEVs and SEVs: for miRNAs in SEVs the percentage was between 18.2–61.5% and in LEVs 25.2–46.2% (Table 2). Although there is some overlap between the two EVs subtypes, there is a significant difference that may justify, as we already described for dimension, protein and lipid loading [28,29,30], the different functions of LEVs and SEVs in plasma of ND patients. Conley et al. characterized protein coding transcripts in SEVs and LEVs from breast cancer patients by RNA-Seq and identified a small fraction of transcripts that were expressed at significantly different levels in large oncosomes and exosomes, suggesting they may mediate specialized functions [38]. SEVs are more enriched in deregulated miRNAs compared to LEVs. This is in agreement with the literature demonstrating that SEVs contain primarily small RNA [39]. Moreover, EVs from AD patients transport very few DE miRNAs compared to the other three NDs.

Regarding common deregulated miRNAs in LEVs and SEVs in ALS, our data showed an important deregulation in a small group of specific miRNAs already described in the literature (hsa-miR-206, hsa-miR-205-5p, miR-1-3p, hsa-miR-205-5p, hsa-miR-200b-3p, hsa-miR-200c-3p, hsa-miR-6888-3p, hsa-miR-31-5p, hsa-miR-141-3p, hsa-miR-210-3p). MiR-1 and miR-206 has already been described to be deregulated in ALS patients [40,41,42], while in previous studies miR-141 and miR-200 were reported as related to ALS since they bind a sequence in FUS promoter and in turn have an impact on FUS protein synthesis [43,44]. Additionally, miR-210 is already known to be up-regulated in neurodegenerative diseases [45]. To our knowledge, remaining miRNAs deregulation was never reported as associated to ALS.

We then analyzed deregulated miRNAs among the four NDs by principal component analysis (PCA). We found that deregulated miRNAs cargo of the four NDs was different from CTRs, in particular in SEVs, while in LEVs, the only group that did not overlap with CTRs was ALS. On the other hand, miRNAs split the group of PD patients in two, one overlapped with AD and the other with FTD patients. Additionally, FTD patients showed two subgroups, one overlapping with PD and the other with ALS. Multiple studies [46,47,48] already demonstrated that the observed overlap between FTD and ALS is due to common mechanisms contributing to the onset and development of the disease. Parkinsonism is found in approximately 20–30% of patients in FTLD; in particular, it is frequently observed in familial FTD, with mutations linked to microtubule associated protein Tau (MAPT), progranulin (GRN or PGRN), and chromosome 9 open reading frame 72 (C9ORF72) repeat expansion [49]. From the clinical point of view, in our cohort, only one FTD patient presented parkinsonism, which, however, clustered in the ALS group and did not present any mutation in the canonical genes associated to familial FTD. All FTD patients showed variable amounts of Tau, βamyloid 1-42 and only two patients presented ALS-FTD disease (Table 3).

Common miRNAs in the four NDs were different between the two groups of EVs. Enriched pathways of common miRNAs found with MiRWalk showed pathway like Ubiquitin mediated proteolysis, MAPK signaling pathway, Toll-like receptor signaling pathways for SEVs and TGF-beta signaling pathway, Neurotrophin signaling pathway, MAPK signaling pathway, Glycosphingolipid biosynthesis, Ras signaling pathway for LEVs. Common miRNAs both in LEVs and SEVs are involved in signal transduction. For example, it is known that Phosphorylated Akt expression is augmented in AD and PD patients [50]. However, although recent reports have implicated EVs in intercellular signaling [51], their influence in modulating signaling pathways in NDs has not been clearly understood.

For the common enriched pathway of SEVs in the four NDs, it is known that many NDs are related to inflammation, which can increase cell injury and cause neuronal death. Toll like receptors (TLR) are innate immune receptors that, when activated, can induce the downstream signal molecules through the MyD88-dependent and TRIF signal adaptor proteins, which activate downstream kinases including IkB kinases and MAP kinases. In general, TLRs are expressed in the CNS, in neurons (TLR3, 4, 7, and 9), in human oligodendrocytes (TLR2), in human astrocytes (TLR3-4), and in human microglia (TLR1-4). Activation of both endosomal and plasma membrane receptors like TLRs can activate microglia and control the evolution of neurodegenerative processes [52].

Ubiquitin mediated proteolysis is the process of degradation of a protein via the ubiquitin system [53] and it is one of the common pathways of SEVs from the four NDs that we found. NDs are characterized by intraneuronal inclusions containing ubiquitynated filamentous protein aggregates, given by loss of function or mutations in enzymes of the ubiquitin conjugation/deconjugation pathway [54]. If there is an impairment of the ubiquitin mediated proteolysis, SEVs, which originate in the endocytic pathway, (differently from LEVs, which are shed from the budding of the cell membrane) might be the affected key part of the machinery as already suggested. In fact, the incorporation of ubiquitinated proteins into intraluminal vesicles (ILVs) is controlled through the Endosomal Sorting Complexes Required for Transport, ESCRT complex, key part of SEVs [55].

For LEVs, Neurotrophin signaling pathway, also called nerve growth factor (NGF) family members pathway, has multiple functions in both developing and mature neurons and it is connected to the downstream MAPK and Ras signaling pathway. On activation by BDNF, trkB initiates intracellular signaling through Shc and PLCγ binding sites. The Shc binding site plays major roles in neuronal survival and axonal outgrowth [56]. Another common deregulated pathway is glycosphingolipid biosynthesis pathway. Increasing evidence underlines the activation of ceramide-dependent pro-apoptotic signaling and reduction of neuroprotective S1P in neurodegeneration course. One hypothesis is the link between altered ceramide/S1P and the production, secretion, and aggregation of pathological proteins in NDs. Sphingolipids regulate EVs and the spread or release of neurotoxic proteins and/or regulatory miRNAs between brain cells [57].

The expression profile of some miRNAs has already been described in the brain or in blood of some NDs, but with an opposite regulatory pattern to the one found in EVs of our study.

For example, miR-133b expression is down-regulated in the midbrain of PD patients and in an animal model of PD, while in SEVs is up-regulated [58]. miR-4781–3p was instead found up-regulated in blood of AD patients by RNAseq, while in SEVs of the four NDs, this miRNA is down-regulated [59]. miR-323–3p associated with inflammatory responses, has been proposed as a target for therapy in AD and it is found down-regulated in SEVs [60]. Some of these miRNAs have not been found in neurodegeneration, and some only in one of the NDs studied in this work. Nevertheless, this study was carried out on a limited number of patients and controls; moreover, healthy donors age matched only two diseases of the four examined. These limitations should be overcome with further studies in which specific miRNAs and pathways will be investigated for each disease.

## 4. Materials and Methods

### 4.1. Study Subjects

Participants were recruited at the IRCCS Mondino Foundation, Pavia (Italy). Plasma isolated from 6 AD, 9 PD, 6 sporadic ALS (SALS), 9 FTD patients were used (Table 4). All patients were screened for mutations using a customized panel of 176 genes associated to neurodegenerative and neuromuscular diseases by Next Generation Sequencing (Sure Select QXT Target Enrichment, Agilent Technology, Santa Clara, CA, USA). ALS and FTD patients were screened for C9orf72 using the FastStart Taq DNA Polymerase Kit (Roche, Basel, Switzerland).

Diagnosis of AD was based on criteria expressed by Aging-Alzheimer’s Association workgroups [61]. For PD and FTD patients Movement Disorder Society (MDS) clinical diagnostic criteria were used [62,63]. ALS diagnosis was made according to the revised El Escorial Criteria [64].

A total of six age-matched healthy volunteers free from any pharmacological treatment were recruited at the Immunohematological and Transfusional Service IRCCS Foundation “San Matteo”, Pavia (Italy) and used as healthy controls (CTRs). All the subjects were assayed to rule out the presence of inflammatory diseases by white blood cell counts and subjects with WBCs >11 × 10^9^ were excluded from the study. Patients’ characteristics are reported in Table 4, Table 5, Table 6 and Table 7.

### 4.2. LEVs and SEVs Isolation

Venous blood (15 mL) was collected in sodium citrate tubes from all patients and controls and processed as previously described [28,29,30]. Briefly, platelet-free plasma was centrifuged at 20,000× *g* for 1 h. The pellet was washed in 0.2 µm filter filtered 1X PBS (Sigma-Aldrich, Milan, Italy). The supernatant of LEVs was filtered through a 0.2 µm filter and spun in an Optima MAX-TL Ultracentrifuge at 100,000× *g* for 1 h at 4 °C and SEVS pellet was washed with 1 mL of filtered 1X PBS. Western blot analysis for LEVs markers (Annexin V, Abcam, Cambridge, UK) and for SEVs markers (Alix-Abcam, Cambridge, UK), Transmission Electron Microscopy (TEM) and Nanoparticle-tracking analysis (NTA) were run to confirm LEVs and SEVs purity as we previously described [28,29].

### 4.3. RNA Extraction

RNA was extracted from LEVs and SEVs fractions using Qiagen miRNeasy Mini kit (Qiagen, Hilden, Germany) according to the manufacturer’s instructions.

### 4.4. RNA Libraries Preparations

Small RNA libraries were constructed with NEBNext^®^ kit (New England Biolabs, Ipswich, MA, USA). Individually-barcoded libraries were mixed. Pools were size selected on Novex 10% TBE gels (Life Technologies, Carlsbad, CA, USA) to enrich for miRNAs fraction. Sequencing (75 nts single-end) was performed on Illumina NextSeq500 (Illumina, San Diego, CA, USA).

### 4.5. Bioinformatic Data Analysis

The raw bcl files were converted into demultiplexed fastq files with bcl2fastq (Illumina, San Diego, CA, USA) implemented in docker4seq package [65]. For the row count analysis, only transcripts with counts above five were considered. No relevant difference between count in SEVs and LEVs in four diseases emerged.

Quantification of miRNAs was done as described in the literature [66]. The workflow, including quality control filter, trimming of adapters, reads mapping against miRNA precursors, is implemented in docker4seq package [65,66]. Differential expression analysis was performed with the R package DESeq2, implemented in docker4seq package. We imposed a minimum |Log2FC| of 1 and a FDR lower than 0.1 as thresholds to detect differentially expressed miRNAs.

In order to understand common miRNAs of the four diseases, we calculated the intersection of deregulated miRNAs compared to CTRs with http://bioinformatics.psb.ugent.be/webtools/Venn/ accessed on 11 December 2019). The datasets generated and analysed during the current study are available in the NCBI GEO repository [GSE155700].

### 4.6. Pathways Analysis

miRNA-targets analysis was done with miRWalk web tool (http://mirwalk.umm.uni-heidelberg.de, accessed on 11 December 2019).

In addition, Ingenuity pathway analysis (IPA, v. 2019 summer release, Qiagen, Germany) was performed to identify miRNAs and biological networks associated with the differentially expressed circulating miRNAs in SEVs and LEVs in each of the neurodegenerative disorders.

## 5. Conclusions

We found different deregulation of miRNAs between SEVs and LEVs from plasma of patients in four NDs.

We found a common signature of miRNAs in SEVs and LEVs among the four NDs and those miRNAs are involved in pathways already known in neurodegeneration.

For each NDs, we also found new DE miRNAs involved in different pathways and this might give a specificity to the role of SEVs and LEVs in the spreading/protection of each disease.

## Figures and Tables

**Figure 1 ijms-22-02737-f001:**
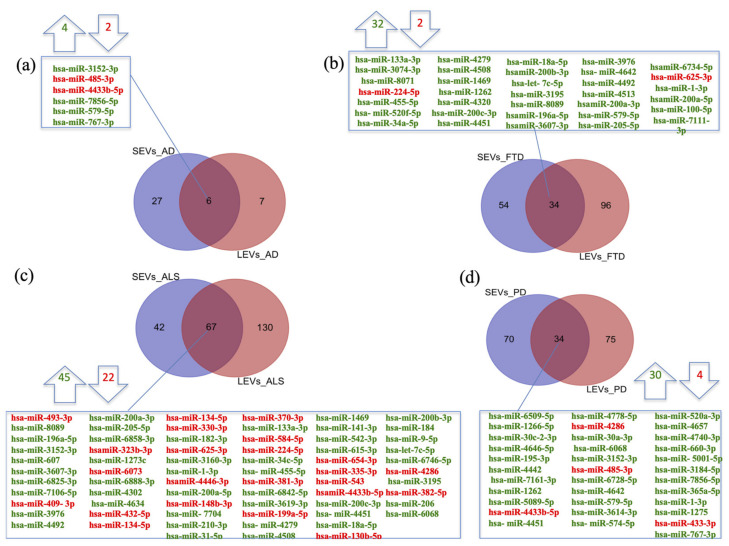
Common packaging of deregulated miRNAs into SEVs and LEVs of NDs. (**a**) In AD, of the 33 miRNAs found in SEVs and 13 in LEVs, 6 distribute to both, 4 up-regulated-in green and 2 down-regulated-in red; (**b**) In FTD, of the 88 miRNAs in SEVs and 130 in LEVs, 34 were in common (32 up-regulated and 2 down-regulated); (**c**) for ALS, of the 109 miRNAs in SEVs and 197 in LEVs, 67 were in common, 45 up-regulated and 22 down-regulated; (**d**) in PD, of the 104 miRNAs found in SEVs and 109 in LEVs, 34 distribute to both, 30 up-regulated and 4 down-regulated. Differential miRNA expression analysis by DESeq2 (log2FC > 1, *p*-value < 0.05). AD = Alzheimer Disease; FTD = Fronto-Temporal Dementia; ALS = Amyotrophic Lateral Sclerosis; PD = Parkinson Disease; CTRs = controls; SEVs = small extracellular vesicles; LEVs = large extracellular vesicles.

**Figure 2 ijms-22-02737-f002:**
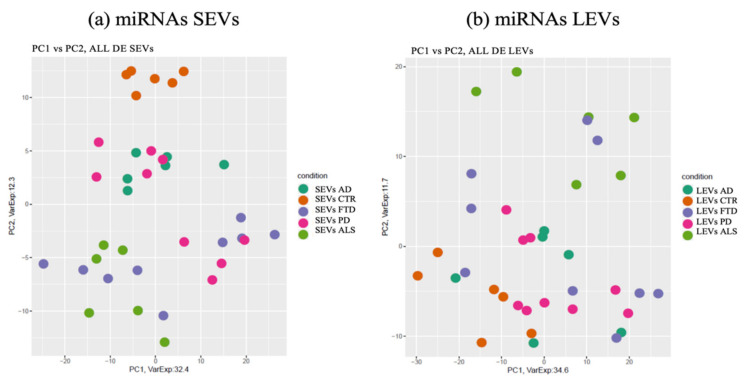
Principal component Analysis (PCA) of miRNAs differentially expressed in SEVs (**a**) and LEVs (**b**) of ALS, FTD, AD and PD patients and healthy controls (CTRs). PCA is performed using as predictors all the miRNAs identified as differentially expressed in at least one disease in the comparison of each disease to the control state. Each dot represents a sample and each color represents a disease. AD = Alzheimer Disease; FTD = Fronto-Temporal Dementia; ALS = Amyotrophic Lateral Sclerosis; PD = Parkinson Disease; CTRs = controls; SEVs = small extracellular vesicles; LEVs = large extracellular vesicles; miRNA = microRNA.

**Figure 3 ijms-22-02737-f003:**
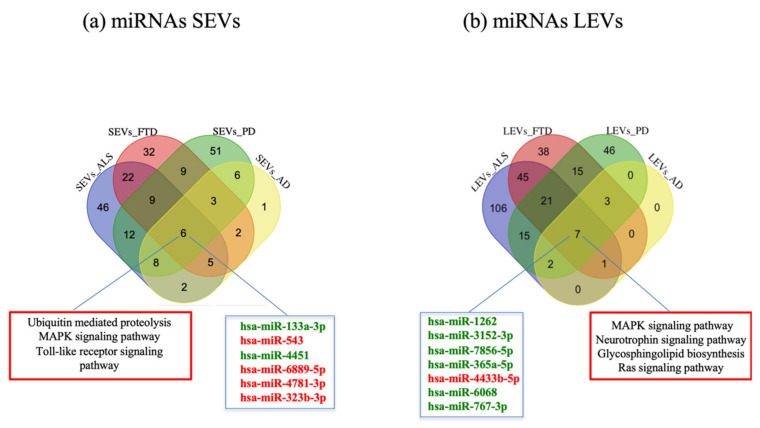
Venn diagram showing numbers of common and unique miRNA and RNA in SEVs (**a**) and LEVs (**b**) from plasma of AD, FTD, ALS and PD patients. Common miRNAs and pathways are listed. Differential miRNA expression analysis by edgeR (log2FC > 1, *p*-value < 0.05). AD = Alzheimer Disease; FTD = Fronto-Temporal Dementia; ALS = Amyotrophic Lateral Sclerosis; PD = Parkinson Disease; CTRs = controls; SEVs = small extracellular vesicles; LEVs = large extracellular vesicles; miRNA = microRNA.

**Figure 4 ijms-22-02737-f004:**
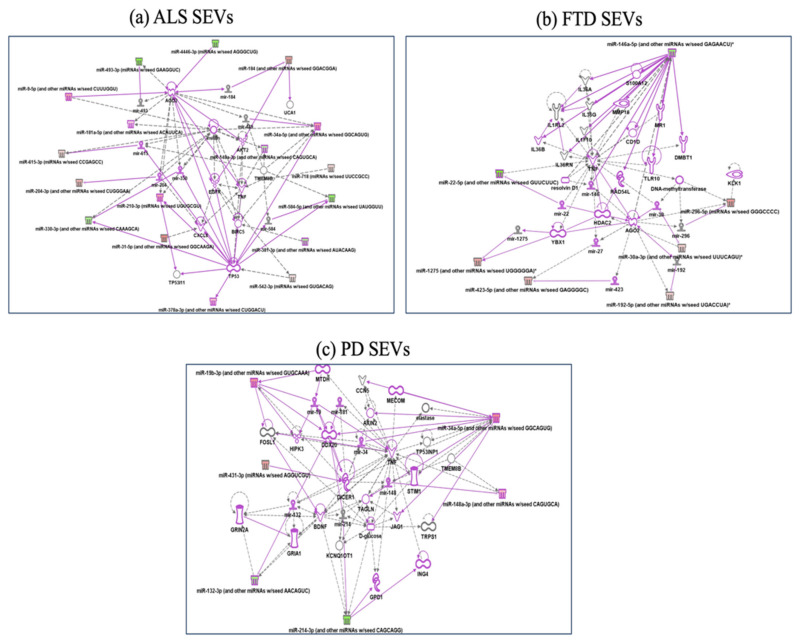
IPA networks among deregulated miRNAs in SEVs from ALS (**a**), FTD (**b**), PD (**c**). For ALS miRNAs involvement in regulation of DDX58/IFIH1, through miR31-5p and miR615-5p (**a**); for FTD IL1F10 and TLR10 regulation by miR146a-5p (**b**); for PD, regulation of GRIA1, Glutamate receptors, by miR-132-3p, involvement of JAG1 and TNF regulated by miR-34 (**c**). Pink color indicates activation while green color indicate suppression. No pathway could be calculated for AD disease for the few miRNAs targets. FTD = Fronto-Temporal Dementia; ALS = Amyotrophic Lateral Sclerosis; PD = Parkinson Disease; CTRs = controls; SEVs = small extracellular vesicles.

**Figure 5 ijms-22-02737-f005:**
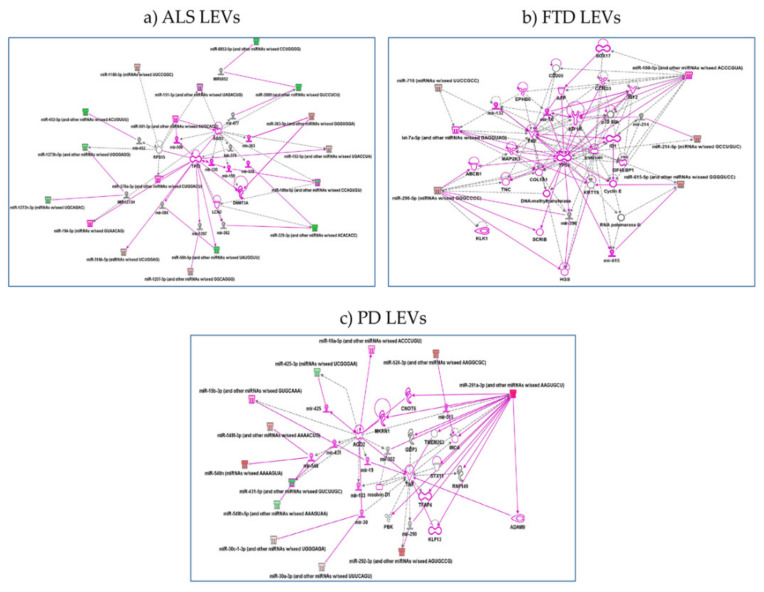
IPA networks among deregulated miRNAs in LEVs from ALS (**a**), FTD (**b**), PD (**c**). (**a**) for ALS, regulation of p53 by mir-379, 378, 584-5p, 1207 emerged or hsa-miR-199a-5p and hsa-miR-329a-5p regulate LCN2, lipocalin2; (**b**) for FTD, regulation of genes like TP53 by miR-296 and RNA Polymerase II by miR-615 as well as Cyclin E regulated miR-615-5p; (**c**) for LEVs from PD patients, regulation of ADAM9 by up-regulation of miR-291. Pink color indicates activation while green color indicate suppression. FTD = Fronto-Temporal Dementia; ALS = Amyotrophic Lateral Sclerosis; PD = Parkinson Disease; LEVs = large extracellular vesicles.

**Table 1 ijms-22-02737-t001:** Statistically significant number of differentially expressed miRNAs in SEVs and LEVs from ALS, FTD, AD, PD patients. Up-regulated transcripts, down-regulated transcripts and total compared to CTRs were reported. Transcripts were considered as differentially expressed when |log2(disease sample/healthy controls)| ≥ 1 and a FDR ≤ 0.1.

miRNAs	AD		FTD		ALS		PD	
	*SEVs*	*LEVs*	*SEVs*	*LEVs*	*SEVs*	*LEVs*	*SEVs*	*LEVs*
up-regulated	17	10	49	113	80	128	85	87
down-regulated	16	3	39	17	29	69	19	22
Total	33	13	88	130	109	197	104	109

AD = Alzheimer Disease; FTD = Fronto-Temporal Dementia; ALS = Amyotrophic Lateral Sclerosis; PD = Parkinson Disease; CTRs = controls; SEVs = small extracellular vesicles; LEVs = large extracellular vesicles; miRNA = microRNA; FDR = False Discovery Rate.

**Table 2 ijms-22-02737-t002:** Percentage of common miRNAs in SEVs and LEVs in AD, FTD, ALS and PD.

miRNA		
NDs	Common miRNAs in SEVs and LEVs (%)	
	*SEVs*	*LEVs*
AD	18.2	46.2
FTD	38.6	25.2
ALS	61.5	34
PD	32.7	31.2

ND = Neurodegenerative Disease; AD = Alzheimer Disease; FTD = Fronto-Temporal Dementia; ALS = Amyotrophic Lateral Sclerosis; PD = Parkinson Disease; CTRs = controls; SEVs = small extracellular vesicles; LEVs = large extracellular vesicles; miRNAs = micro RNAs.

**Table 3 ijms-22-02737-t003:** Clinical characteristics of FTD subjects.

Patients	Sex	Age	Education (years)	Disease Duration (months)	Familiarity	cMMSE ^†^	Tau ^††^	pTau ^†††^	Aβ ^††††^ 1-42
FTD1	M	54	8	24	No	8,97	<10	10	683
FTD2	F	57	8	36	No	15.74	419	43	616
FTD3	M	70	8	36	No	18	NA	NA	NA
FTD4	M	62	8	34	No	17	164	42	1312
FTD5	M	52	8	34	No	17.97	195	42	1378
FTD6	M	69	11	36	No	21	NA	NA	NA
FTD7	M	56	13	60	No	17.99	<10	7	360
FTD8	M	49	5	12	No	18.31	799	68	1780
FTD9	F	75	8	44	No	17	NA	NA	NA

FTD = Fronto-Temporal Dementia; cMMSE = correct Mini Mental State Examination, i.e., the MMSE corrected for education; Aβ = Amyloid beta; NA = not available; M = male; F = female. ^†^ cMMSE = normal values > 24–30. ^††^ Tau = normal values <375 pg/mL. ^†††^ pTau = normal values < 52 pg/mL. ^††††^ Aβ 1-42 = normal values > 550 pg/mL.

**Table 4 ijms-22-02737-t004:** Baseline characteristics of recruited subjects for this study. Age is reported as mean ± SD. The percentage of male and female subjects is also indicated.

Column Title	CTRs	AD	FTD	ALS	PD
Recruited subjects	6	6	9	6	9
Age (mean ± SD)	55 ± 5.2	77 ± 3.7	60 ± 6.7	72 ± 6.3	69 ± 3.6
Males %	43%	50%	78%	50%	60%
Females %	67%	50%	22%	50%	40%

CTRs = controls; AD = Alzheimer Disease; FTD = Fronto-Temporal Dementia; ALS = Amyotrophic Lateral Sclerosis; PD = Parkinson Disease; SD = standard deviation.

**Table 5 ijms-22-02737-t005:** Clinical characteristics of AD subjects.

Patients	Sex	Age	Education (years)	Disease Duration (months)	Familiarity	Onset	Imaging ^†^	MMSE ^††^	CDR ^†††^
AD1	M	78	5	50	No	Amnesic	1.3	15	2
AD2	M	70	8	48	Yes	Amnesic	1	19	1.5
AD3	F	79	5	48	No	Amnesic	2	20	1
AD4	M	77	5	40	No	Amnesic	1	13	2
AD5	F	82	5	38	Yes	Amnesic	1	18	1
AD6	F	76	5	42	Yes	Amnesic	2.3	17	2

AD = Alzheimer Disease; MMSE = Mini Mental State Examination; CDR = Clinical Dementia Rating; M = male; F = female. ^†^ Imaging: 1 temporal atrophy; 2 diffuse atrophy; 3 vascular suffering. ^††^ MMSE: normal values > 24–30. ^†††^ CDR: 1 mild; 2 moderate; 3 severe.

**Table 6 ijms-22-02737-t006:** Clinical characteristics of ALS subjects.

Patients	Sex	Age	Education (years)	Disease Duration (months)	Familiarity	Site of Onset	ALSFRS ^†^	Therapy (Riluzole)
ALS1	F	69	13	19	No	Spinal	9	
ALS2	F	70	NA	52	No	Spinal	29	19
ALS3	F	69	NA	11	No	Spinal	34	20
ALS4	M	71	5	24	No	Spinal	46	13
ALS5	M	72	13	67	No	Spinal	28	18
ALS6	M	67	5	28	No	Spinal	40	17

ALS = Amyotrophic Lateral Sclerosis; ALSFRS = ALS Functional Rating Scale; NA = not available; M = male; F = female. ^†^ ALSFRS: minimum score: 0; maximum score: 40. The higher the score the more function is retained.

**Table 7 ijms-22-02737-t007:** Clinical characteristics of PD subjects.

Patients	Sex	Age	Education (years)	Disease Duration (years)	Familiarity	Affected Side	UPDRS III ^†^	Therapy LEDD ^††^	Disorders of Behaviour
PD1	F	70	NA	8	No	Right	12	350	No
PD2	M	73	NA	7	Yes	Right	10	320	ICD
PD3	M	68	8	19	No	Left	44	800	ICD
PD4	M	69	NA	13	Yes	Right	18	500	No
PD5	F	73	8	10	Yes	Right	17	1000	No
PD6	M	76	NA	7	No	Right	18	650	No
PD7	F	68	NA	6	No	Right	20	600	No
PD8	M	76	8	13	Yes	Left	35	690	No
PD9	F	73	NA	13	Yes	Left	15	350	ICD

PD = Parkinson Disease; ^†^ UPDRS III = Unified Parkinson’s disease Rating Scale-motor part. Patients where tested during the ON phase; ^††^ Therapy LEDD: levodopa equivalent daily dose (LEDD), mg/day; NA = not available; ICD = Impulse Control Disorders.

## Data Availability

The datasets generated and analysed during the current study are available in the NCBI GEO repository [GSE155700].

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
