# Peer review of "Different miRNA Profiles in Plasma Derived Small and Large Extracellular Vesicles from Patients with Neurodegenerative Diseases"

_ijms, 2021, doi:10.3390/ijms22052737_

Round 1

Reviewer 1 Report

Sproviero et al report here the identification of differentially expressed (DE) miRNAs in circulating LEVs and SEVs from patients with NDs: AD, PD, ALS, and FTD in comparison to healthy individuals. Common DE miRNAs among these 4 NDs have been identified. Pathway alterations or enrichment associated with these DE miRNAs have been characterized.

This is a much-needed study in the field of ND. DE miRNAs presented in the individual NDs and shared in these NDs have biomarker potentials for early diagnosis and may possess causative roles to these NDs. This work deserves a publication.

Nonetheless, some minor revisions are suggested.

This research focuses on miRNA but not “RNA profiles” in general. The title should be revised to reflect this.

The authors frequently used the term “robust biomarkers”, indicating these DE miRNAs being robust biomarkers. The current study only indicates their biomarker potential, which was not directly examined, let alone the robustness. Please rephrase this statement.

The number of patients was from 6-9 for Ctrl and individual NDs. Please acknowledge this limitation. Also, age of CTRs was not well match at least to AD and ALS patients; this limitation should be acknowledged.

Line 141, please use the disease term rather than “violet”.

Table S2 – Please only list those pathways with p value < 0.05 and define “Pop” and “BH”. Please cite Table S2 in the paragraphs started at lines 176. 187, and 211 respectively. Figure 4a should be referred in manuscript.

Line 250 – Should “mRNA” be miRNA?

Author Response

Dear Reviewer,

thank you for your comments. Below I have listed my answers. All the changes have been made in the document and underlined in yellow:

1) I have changed the title in the attached word file;

2) I have removed robust from each sentence;

3) I have added a comment in the discussion "Nevertheless, this study was carried out on a limited number of patients and controls; moreover, healthy donors age matched only two diseases of the four examined. These limitations should be overcome with further studies in which specific miRNAs and pathways will be investigated for each disease.  ";

4) I have changed with the disease name;

5) I have changed the table and added the acronyms for BH and Pop Total; I have added Figure 4a and table S2 to the text;

6) I have corrected mRNA with miRNA.

Reviewer 2 Report

In this qualitative study, the authors investigate common and different miRNA present in LEVs and SEVs of Alzheimer’s Disease (AD), Parkinson’s disease (PD), Amyotrophic Lateral Sclerosis (ALS), and Fronto-Temporal Dementia (FTD) patients. LEVs and SEVs were isolated from plasma of patients and healthy volunteers (CTR) by filtration and ultracentrifugation and RNA was extracted. miRNA libraries were carried out by Next Generation Sequencing (NGS). The miRNA in the LEVs and the SEVs are involved in different pathways and this might give specificity to their role in the spreading of the disease. 

Overall, this qualitative study is very well planned. The results sufficiently support the hypothesis. 
There are only a few comments that the authors could be considered for improving the manuscript.

In table 3, the authors, recruitment of 9 patients with PD disease is it impossible that 50% are male and 50% are female. 

The quality of WB showed in Supp file 1 should be better

Author Response

Dear Reviewer,

many thanks for your comments. I will reply in points to your minor revisions:

1) In table 3, I have modified the percentage of male and female  PD patients to 60%  and 40%; 

2) I have improved the quality of supplementary figure 1 as you can see in the word file attached
